# Machine Learning Methods for Cancer Classification Using Gene Expression Data: A Review

**DOI:** 10.3390/bioengineering10020173

**Published:** 2023-01-28

**Authors:** Fadi Alharbi, Aleksandar Vakanski

**Affiliations:** Department of Computer Science, University of Idaho, Moscow, ID 83844, USA

**Keywords:** gene expression analysis, machine learning, cancer classification

## Abstract

Cancer is a term that denotes a group of diseases caused by the abnormal growth of cells that can spread in different parts of the body. According to the World Health Organization (WHO), cancer is the second major cause of death after cardiovascular diseases. Gene expression can play a fundamental role in the early detection of cancer, as it is indicative of the biochemical processes in tissue and cells, as well as the genetic characteristics of an organism. Deoxyribonucleic acid (DNA) microarrays and ribonucleic acid (RNA)-sequencing methods for gene expression data allow quantifying the expression levels of genes and produce valuable data for computational analysis. This study reviews recent progress in gene expression analysis for cancer classification using machine learning methods. Both conventional and deep learning-based approaches are reviewed, with an emphasis on the application of deep learning models due to their comparative advantages for identifying gene patterns that are distinctive for various types of cancers. Relevant works that employ the most commonly used deep neural network architectures are covered, including multi-layer perceptrons, as well as convolutional, recurrent, graph, and transformer networks. This survey also presents an overview of the data collection methods for gene expression analysis and lists important datasets that are commonly used for supervised machine learning for this task. Furthermore, we review pertinent techniques for feature engineering and data preprocessing that are typically used to handle the high dimensionality of gene expression data, caused by a large number of genes present in data samples. The paper concludes with a discussion of future research directions for machine learning-based gene expression analysis for cancer classification.

## 1. Introduction

Cancer describes a class of diseases in which malignant cells form inside the human body due to genetic change. These cells divide indiscriminately upon development, extend throughout the organs, and in many cases, they can result in loss of life. Cancer is the second leading cause of mortality globally after cardiovascular illnesses [1]. Recently, gene expression analysis has emerged as an important means for addressing the fundamental challenges associated with cancer diagnosis and drug discovery [2,3]. Gene expression analysis also provides insights into the contribution of different genes to cancer initiation and progression. Consequently, changes in gene expression can be used as markers for the early detection of cancer, and for identifying targets for drug development. Such approaches can open up the possibility of healthcare that is more personalized, preventative, and predictive [4].

Gene expression is the process by which the information contained in DNA is transformed into instructions for making proteins or other molecules. It involves the transcription of DNA into messenger RNA (mRNA), followed by a translation into proteins. Gene expression analysis is employed to assess the order of genetic alterations occurring under certain conditions, in tissue or a single cell [5]. It involves measuring the number of DNA transcripts present in a sample tissue or cells to obtain information about which genes are expressed and to what levels. A component of gene expression quantification is a comparison of the sequenced reads related to the number of base pairs sequenced from a DNA fragment to a recognized genomic or transcriptome source. The precision of the quantification depends on the sequenced reads having sufficient distinctive information to allow applying bioinformatics algorithms to correlate the reads to the appropriate genes. Prevalent methods for estimating gene expression include DNA microarrays and next-generation sequencing (NGS) methods. The DNA microarray method employs a two-dimensional array with microscopic spots to which short sequences or genes bind to known DNA molecules through a hybridization process. NGS methods of massively parallel sequencing offer extraordinarily high-throughput analysis, scalability, and speed, and they have been used to determine the nucleotide sequence of a full genome, or of a single DNA or RNA segment [6,7]. RNA-sequencing, also known as RNA-Seq, is an NGS method that involves the conversion of RNA molecules into complementary DNA (cDNA) and determining the sequence of nucleotides in the cDNA for gene expression analysis and quantification. Compared to DNA microarrays, RNA-Seq [8,9] provides several advantages, including greater specificity and resolution, increased sensitivity to differential expression, and greater dynamic range. RNA-Seq can also be used to examine the transcriptome for any species to determine the amount of RNA at a specific time.

Gene expression analysis requires implementing computational methods for understanding how genes are regulated or their role in the functioning of tissues and cells. machine learning (ML)-based approaches have been frequently used to obtain insights related to how variations in genes and regulatory regions result in phenotypic changes, such as traits, wellness, and health [10,11]. Whereas early computational methods for gene expression analysis typically relied on conventional ML approaches, such as Decision Trees and Support Vector Machines, in the past ten years, deep learning (DL)-based methods for forecasting the structure and function of genomic components—such as promoters, enhancers, or gene sequence levels—have grown in prominence [12,13].

An important component of computational methods for gene expression analysis is feature engineering, employed to handle the challenge of high dimensionality and the relatively small number of samples in gene expression data. This study provides an overview of the feature engineering techniques in gene expression analysis, classified into filter, wrapper, and embedded methods [14]. Filter methods remove irrelevant and redundant data features based on quantifying the relationship between each feature and the target predicted variable. Wrapper methods employ a classification algorithm for evaluating the importance of data features, where the classifier is wrapped in a search algorithm to discover the best subset of data features. Embedded approaches [15,16] identify important features that enhance the performance of a classification algorithm by embedding the feature engineering technique into the learning stage of the classifier. In general, filter methods are characterized by fast processing and lower computational complexity. In contrast, wrapper and embedded methods typically extract more relevant data features that contribute to improved performance of the corresponding classification method.

In the published literature, various architectures of deep neural networks (NN) were applied for cancer classification using gene expression data, including fully connected neural networks (also known as multi-layer perceptron NN, or MLP), convolutional neural networks (CNN), recurrent neural networks (RNN), graph neural networks (GNN), and transformers neural networks (TNN). MLP networks have connections from each neuron to all neurons in the previous and the next layers. For analysis of gene expression data, the input layer in MLP receives the gene expression profiles with each probe received by one neuron. The output layer of the MLP returns the class probabilities of the gene expression sample [17]. CNN models were initially designed for processing multidimensional array data (images in particular), having two-dimensional convolutional filters as processing units for learning hierarchical data representations. Subsequently, several works transformed gene expression data into two-dimensional image-like arrays with rows and columns [18] that were used as inputs to the network. Due to the ability of CNN models to capture local spatial relations in input data, they typically produced better classification performance for gene expression analysis in comparison to MLP methods. In addition, prior works also applied one-dimensional CNN, where each row in gene expression data is fed directly as input to networks having layers that apply one-dimensional convolutional filters. CNN models have generally been some of the best-performing DL models for gene expression analysis. RNN models employ network architectures with recurrent connections, designed for modeling sequential data. A state vector is employed as an intermediate process that combines the information of the current input in the sequence and stored previous values to produce the output. These properties make RNN suitable for capturing correlations in sequences of gene expression data as a source of information regarding the biological processes underpinning cancer development [19,20]. Among the limitations of RNN are the increased computational cost, and they are more susceptible to overfitting in the small data regime, in comparison to CNN. GNN models employ an architecture designed to learn graph representations of data features via a set of graph nodes and edges. These models transform gene expression data into a graph representation and use a gene expression topology to understand the correlations between the different genes [21]. This capacity for learning graph-structured representations renders great potential for GNN in future gene expression analysis, as demonstrated in recent works. TNN models use a network architecture that applies the self-attention mechanism for learning long-range dependencies in sequential data. This property makes TNN well-suited for identifying correlations in gene expression analysis, and subsequently, models have been employed in previous studies. Unlike related sequence models such as RNN, TNN allows parallelization of the input samples for model training, which results in faster processing of long sequences. Additionally, several studies designed hybrid network architectures, such as TNN models with 1D convolutional layers, constructed for extracting gene information shared between cancer types without the need for feature selection [22]. Additionally, transfer learning, which refers to a set of techniques for transferring information from one model trained with a large dataset to another, has been used to tackle the problem of small training datasets and the high dimensionality of gene expression data [23,24].

Despite the recent progress in ML-based cancer classification using gene expression data, various challenges remain to be addressed. Specifically, gene expression datasets typically contain a small sample size, where each sample has a relatively large number of dimensions, i.e., the number of genes. To handle this challenge, ML methods usually rely on feature-engineering techniques for removing redundant information and selecting an optimal set of features for classification. Although conventional ML approaches are more dependent on efficient feature engineering and data preprocessing, prior works have also leveraged feature engineering techniques in the workflows of DL models to improve performance. In addition, transfer learning techniques were also employed to overcome the problem of model training in the small data regime. DL-based methods have generally outperformed conventional ML methods, and it can be expected that most future models for gene expression analysis will be based on DL networks. Currently, several approaches that employed MLP or CNN networks in combination with efficient feature engineering and transfer learning techniques have achieved test accuracies upwards of 90%. However, the performance of current methods is sensitive to various parameters, and further improvements are required for the generalization and robustness of the methods. Other limitations of existing approaches include the lack of interpretability and limited integration with other data types and modalities.

Numerous review papers in the published literature have overviewed the advances in computational approaches for gene expression analysis. The most relevant papers that are the closest to this review and were published in the last three years are listed in Table 1. The table provides comparative information about the review papers, i.e., whether they covered conventional ML approaches, feature-engineering techniques, DL approaches, and the type of gene expression data used by the reviewed techniques. For instance, several papers reviewed only conventional ML approaches for gene expression analysis. Similarly, some of the reviews concentrated solely on feature-engineering techniques, or other aspects of gene expression analysis. In addition, many previous studies discussed mainly the computational approaches related to DNA microarray gene expression data. Although there are similarities and overlaps to some of the reviews listed in Table 1 and to other reviews published before 2019, this review provides novel insights that were not covered in prior works. The main contribution of this survey includes a comprehensive overview of the applications of both conventional ML approaches and recent DL approaches for gene expression analysis. Although some of the prior reviews discuss case studies in gene expression analysis using DL architectures such as MLP, CNN, and RNN, no previous review offers a comprehensive discussion of the use of GNN [25,26] and TNN [27,28] architectures for gene expression analysis. On the other hand, GNN and TNN have the potential to become prevalent DL architectures for this task. Furthermore, the focus of this study was on approaches for modeling RNA-Seq gene expression data, being the most dominant data format used for this task in recent years. In addition, this study provides a review of related feature-engineering techniques and datasets for ML-based gene expression analysis, which are not covered in many related review papers.

## 2. Gene Expression Data

Gene expression analysis is the process of identifying the number of transcripts present in a particular cell or tissue type to estimate the level of expressed genes. The branch of science that focuses on the quantitative examination of the transcriptome is transcriptomics. Early computational transcriptomics methods employed Sanger sequencing of expressed sequence tag (EST) libraries. EST libraries represent short fragments of mRNA obtained from a single sequencing procedure carried out on randomly chosen clones from cDNA libraries. Whereas, a cDNA library is a collection of DNA sequences that have been cloned and are complementary to mRNA that has been retrieved from an organism or tissue. Over 45 million EST libraries from approximately 1400 distinct cellular species have been produced to date. Although EST libraries provide a base resolution profile of expressed gene sequences, this technology usually does not contain full-length gene sequences, and subsequently, the methods based on EST libraries were superseded by chemical tag-based techniques, such as Serial Analysis of Gene Expression (SAGE). The SAGE method allows for quantitative and simultaneous analysis of a large number of transcripts in any particular cell system, without prior knowledge of the genes. This method is based on a theoretical calculation that assumes a random nucleotide distribution throughout the genome. The methods of Sanger sequencing of EST libraries and SAGE were succeeded by DNA microarrays and NGS methods—most notably RNA-Seq—for estimating gene expression.

### 2.1. Microarray Data

Microarray data are obtained through a laboratory technique where a DNA sequence is contained in a tool consisting of a two-dimensional array with thousands of microscopic spots. The microarray tools are also known as chips or slides, and each spot on the slide is reserved for a single DNA sequence or gene. The DNA samples bind to the microarray slide through a hybridization process, which is followed by scanning the colors of the spots on the slide to measure the expression of each gene. One row in the microarray data represents the gene expression level, and the columns represent the samples.

Microarrays can be used to identify DNA (as in comparative genomic hybridization) or RNA (often as cDNA following reverse transcription), which may or may not be translated into proteins. Microarray data allow for the comprehension of cellular processes for genome-wide expression profiles related to specific conditions or diseases, such as cancer. Similarly, they provide helpful information used in the pursuit of new pharmaceuticals, in pharmacogenomics, and in the development of effective medications for therapeutic methods.

One of the main advantages of DNA microarrays is that they allow one to measure the expression level of thousands of genes. Microarrays also have limitations, including relatively poor accuracy, precision, and specificity. Another limitation is the high sensitivity of the experimental setup to changes in the temperature of hybridization, the purity and rate of genetic material degradation, and the amplification process, all of which may affect the quantification of gene expression.

### 2.2. RNA-Seq Data

RNA-Sequencing (RNA-Seq) belongs to the NGS methods [41], which are characterized by a capability for rapid profiling, and allow researchers to investigate the transcriptome for any species in determining the presence and amount of RNA at a specific time [42]. This approach has been used to produce millions of sequences from complex RNA samples. RNA-Seq is employed to measure gene expression, examine variations in gene expression over time or due to applied therapies, discover and annotate complete transcripts, examine post-transcriptional modifications, and characterize alternative splicing and polyadenylation. The different applications are based on the capacity to analyze all RNA molecules in a cell or tissue—including protein-coding RNA (mRNA) and non-coding regulatory RNA (miRNA, siRNA) or functional RNA (tRNA, rRNA)—and suitably measure their abundances simultaneously. Other important qualities of RNA-Seq include its high resolution and large dynamic range, which have resulted in a large volume of acquired data and contributed to remarkable advances in transcriptomics research [43]. Due to the above advantages, RNA-Seq has been replacing microarrays for gene expression analysis.

Table 2 presents a comparison between microarray and RNA-Seq data in terms of discovered gene range, different isoforms, resolution, background noise, cost, rare/new transcript, and noncoding RNA. Conclusively, RNA-Seq provides several important advantages when compared to microarray data.

Similarly, single-cell RNA sequencing (scRNA-Seq) [44] allows the profiling of the entire transcriptome for a large number of different types of individual cells. This results in a high-throughput analysis with much larger datasets than conventional RNS-Seq. It allows researchers to determine which genes are expressed in a heterogeneous sample at the single-cell level, in what quantities, and across thousands of cells.

### 2.3. RNA-Seq Data Collection

To obtain RNA-Seq data, the extracted RNA is first converted into cDNA, and next cDNA libraries are prepared for sequencing, and are sequenced on an NGS platform. The sequencing process involves isolating and purifying mRNA molecules from the data. cDNA is obtained by reverse transcription of mRNA using reverse transcriptase enzyme. cDNA libraries are prepared by amplifying the cDNA fragments, and afterwards, NGS platforms are used to analyze the resultant short-read sequences and estimate the gene expression levels. The procedure depends on a number of experimental factors, including the utilization of biological and technical replicates, level of sequencing, and target transcriptome coverage. In certain instances, these experimental options will not significantly affect the quality of RNA-Seq data. Still, thoughtful experimental design is often required, with a focus on striking a balance between high-quality outcomes, time, and financial expenditure.

### 2.4. Gene Expression Datasets

The research community has dedicated significant efforts to collect, organize, and integrate various types of gene expression data. Table 3 lists gene expression datasets, including RNA-Seq and microarray data based on human tissue. The datasets are open-source, easily accessible, and widely used for cancer classification and its related tasks.

## 3. Feature Engineering

Feature engineering is the process of turning raw data into features to emphasize relevant information in the data and/or enhance the data analytics capability of machine learning models. Feature engineering can select relevant features or produce novel features for both supervised and unsupervised learning to streamline and accelerate data transformations while simultaneously increasing the predictive potential of computational methods. Such techniques have been used to extract marker genes that influence the discriminative capabilities of ML models [59,60] by evaluating and appraising which genomic features are not redundant and should be prioritized [61]. In RNA-Seq data, characterized by a large number of genes in comparison to the number of samples, feature selection is employed to select a group of genes that best represent the dataset structure in a lower-dimensional space and increase the signal-to-noise ratio.

Existing algorithms for feature engineering of gene expression data can be categorized into three major groups: filter, wrapper, and embedded methods. The information flow in the three groups of feature engineering methods is depicted in Figure 1.

### 3.1. Filter Methods

Filter methods include feature engineering techniques that filter out data features that are unlikely to contribute to the performance of a predictive model used in gene expression analysis [63]. Filter methods are commonly used as a preprocessing step to rate the importance of all input features, typically by estimating a relevance score for ranking the genes and using a threshold scheme for selecting the relevant genes for further processing [64]. To this end, filtering techniques assign weights related to the intrinsic qualities of the data and evaluate the discriminative capability of the features, which are afterwards utilized to retain only the highest-ranking traits and reject the lower-ranking features. For instance, the Grouping Genetic Algorithm (GGA) was used to tackle the problem of grouping features with the greatest diversity in RNA-Seq data. It has been employed for the classification of an unbalanced database of RNA-Seq samples for gene expression analysis with different forms of cancer. An advantage of the filter class of feature engineering methods is that it is fast and computationally inexpensive, and therefore can be applied to large-scale RNA-Seq datasets. Table 4 lists frequently used filter methods for feature engineering in RNA-Seq data and the performance of the associated predictive models.

### 3.2. Wrapper Methods

Wrapper methods evaluate the significance of data features by employing a classification algorithm. Wrapper methods interact with the classifier to identify a subset of attributes that are optimal for model predictions, i.e., the attributions of various gene subsets are assessed in an initial phase using the classifier to be employed, and afterwards the classifier is re-trained using the genes with high importance. The criterion for selecting a subset of features with wrapper methods is the performance of the learning algorithm. In contrast, the learning algorithm is wrapped in a search algorithm that will discover the best subset of features. In other words, wrapper methods evaluate the subsets of features using the learning algorithm as a black box and the learning algorithm performance as the objective function. Wrapper methods can be categorized as deterministic or randomized search algorithms. Classification models that have been used with wrapper methods include k-Nearest Neighbors [72], Random Forests [73,76], Support Vector Machines [75,77], and others. Since this feature-engineering approach requires training and evaluating a multitude of classifiers by considering different subsets of candidate features, wrapper methods are substantially more time-consuming and computationally demanding than filter methods. On the other hand, wrapper methods deliver improved performance.

### 3.3. Embedded Methods

Embedded methods comprise algorithms for feature engineering that consider the configuration of a used classifier to explore the space of hypotheses and feature subsets in search of the optimal subset of features. Embedded methods attempt to combine the benefits of filter and wrapper methods by applying the advantages of these methods to the specifics of a single learning algorithm. Embedded methods generally produce an improved performance in comparison to filter and wrapper methods, as they have increased capacity for dealing with the feature interaction problem. This problem occurs when the interactions of a subset of genes with other genes influence the feature selection process, making it prone to be stuck with a subset of locally optimal features [81].

### 3.4. Hybrid Methods

Several works in the literature have employed a hybrid approach that combines the advantages of the different feature-engineering methods. For instance, filter methods can be employed in an initial step to reduce the overall number of features that are transmitted to the wrapper stage. Afterward, a classifier model is applied to further refine the features and select the final subset of genes [82]. Such methods introduce a tradeoff between computational complexity and performance. In addition, ensemble methods such as Bagging, Boosting Ensembles, and Random Forests were applied as flexible and robust alternatives for handling feature interactions in high-dimensional settings. Because ensemble methods use multiple weak classifiers, e.g., to fit portions of the available training data or portions of the input features, these methods have been shown to reduce overfitting and improve predictive performance in gene expression analysis [83].

### 3.5. Advantages and Disadvantages of Feature Engineering Methods

Table 5 presents the most important benefits and drawbacks of feature-engineering methods in gene expression analysis. The advantages and limitations of filter methods are provided for univariate filters and multivariate filters, where univariate filters evaluate each feature independently, whereas multivariate filters evaluate features from the perspective of other data features. For the wrapper methods, benefits and drawbacks are provided based on prior knowledge about the feature distribution. Deterministic methods are used when random variations of features in the data have a major influence on the predictive model, whereas randomization procedures do not make assumptions about the data distribution or account for random variations of features for gene expression analysis.

## 4. Methods for Gene Expression Analysis

Various ML methods have been used in gene expression analysis to identify potential cancers and provide insights into potential treatment options.

### 4.1. Traditional Machine Learning Methods

Conventional machine learning methods, such as Support Vector Machines (SVM), k-Nearest Neighbor (kNN), Naïve Bayes (NB), Random Forest (RF), and related methods were used in a body of works on early cancer detection [84,85]. For instance, Segal et al. [86] proposed a genome-based SVM strategy for the classification of clear cell sarcoma. The authors employed the Student’s t-test to select a set of 256 genes, which were used to train a linear SVM classifier for distinguishing melanoma and soft tissue sarcoma. The classifier accurately identified 75 out of 76 instances in leave-one-out cross-validation. Further, several traditional ML methods were combined with feature selection methods, such as the work of Zhang et al. [73], who use SVM with recursive feature elimination (RFE) and parameter optimization (PO), hence referred to as SVM–RFE–PO. This approach applied grid search and Partial Swarm Optimization for feature selection, combined with a genetic algorithm for parameter tuning in the feature selection process. Afterwards, the optimal set of salient features was used to train an SVM model for cancer classification. Ram et al. [87] implemented an RF ensemble to extract a set of 273 relevant genes while retaining the predictive capacity of the classifier. Similarly, Hijazi et al. [88] introduced an approach for selecting a group of genes that can best differentiate between cancer subtypes for normal and cancer samples via a two-step feature selection strategy based on an attribute estimate method and a Genetic Algorithm. Although the model achieved high accuracy of 99.89% and 99.40% for two types of cancers from five cancer datasets, the performance decreased for other types of cancer. The Evolutionary Programming-trained Support Vector Machine (EP-SVM) method [88] constructed a probabilistic SVM approach to examine the outputs of binary classifiers using unique class features. Prior works that employed traditional ML methods for gene expression analysis are listed in Table 6. 

In general, ML algorithms have proven to be a powerful tool for detecting hard-to-discern patterns in complex and high-dimensional data across numerous applications. Therefore, they have been well-suited for analysis and classification of gene expression data [89]. However, the performance of conventional ML algorithms highly depends on the quality of supplied features; hence, their performance has relied on the efficiency of the accompanying feature selection methods.

### 4.2. Deep Learning Methods

Deep learning-based methods employ artificial neural networks (NN) with multiple layers of processing units for learning data representations. These methods can learn hierarchical representations in high-dimensional data, which is a key advantage compared to conventional ML algorithms [92]. Consequently, present state-of-the-art methods for gene expression analysis take advantage of their unique capabilities [93]. The most commonly used NN architectures include fully-connected NN (multi-layer perceptron NN), convolutional NN (CNN), recurrent NN (RNN), graph NN (GNN), and transformer NN (TNN) [31].

#### 4.2.1. Multi-Layer Perceptron (MLP) Neural Networks

MLP is a neural network architecture with fully connected layers where each neuron in a hidden layer is connected to all other neurons in the neighboring layers. MLP classifiers have been designed for cancer classification in a line of prior works on gene expression analysis. For instance, Lai et al. [94] designed an MLP network that combined diverse data sources of gene expression and clinical data to successfully predict the overall survival of non-small cell lung cancer (NSCLC) patients. The study integrated 15 biomarkers with clinical data, which were afterward utilized to create an integrative MLP classifier using bimodal learning to predict the 5 year survival status of NSCLC patients and achieved 0.8163 AUC and 75.44% accuracy. Zhang et al. [95] proposed an unsupervised feature learning framework for identifying different properties from gene expression profiles by combining a principal component analysis (PCA) algorithm and an autoencoder MLP model. An ensemble classifier based on the AdaBoost algorithm referred to as PCA-AE-Ada was used to predict clinical outcomes in breast cancer. Gao et al. [96] proposed the Deep Cancer Subtype Classification (DeepCC) approach for supervised cancer classification based on the analysis of functional spectra that indicate the activities of biological pathways. They performed enrichment analysis for each sample and trained a multilayer NN to replace hand-engineered features. The authors achieved a balanced accuracy greater than 90% on breast and colorectal cancer classification. Chandrasekar et al. [97] introduced an MLP-based classification approach to compare the assessment period, categorization correctness, and potential to detect the illness and determine the severity positions of the illness. They focused on obtaining accurate prediction using a small number of gene subsets in predicting cancer disease and providing its severity level. Laplante et al. [98] designed an MLP network for classifying cancers in 20 anatomical areas using miRNA stem-loop cohorts to identify the anatomical site of cancer in 27 cancer types from TCGA. The first layer of the MLP network had 1046 input neurons corresponding to each miRNA in the dataset, and the final layer had 27 neurons representing the cancer types. This approach achieved an average accuracy of 96.9%.

Table 7 lists related works in gene expression analysis based on MLP neural networks. The main advantage of MLP models and related DL methods compared to conventional ML methods is the capacity for extracting representative features in genomic data independently from the implementation of feature selection methods. Among the limitations of MLP classifiers is that the fully connected network architecture is less powerful in identifying long-term correlations in genomic data compared to CNN, RNN, GNN, and TNN. MLP networks also lack the means provided by GNN for identifying graph connections in genomic data.

#### 4.2.2. Recurrent Neural Networks (RNN)

RNN is a subclass of neural networks that introduces recurrent connections between the neuron units, which furnish the network with a memory capability: past observations can be employed to understand the current observation or predict future observations in an input sequence. These characteristics provide RNN with sequential dynamic behavior, making it suitable for processing sequential data and identifying inner relationships and variation tendencies [101,102]. Sahin et al. [103] developed an RNN framework to model a stability mechanism for robust feature selection of microarray datasets. They combined a long short-term memory (LSTM) network with the Artificial Immune Recognition System (AIRS) and achieved 89.6% accuracy. RCO-RNN was introduced by Aher et al. [104] and employed the rider chicken optimization (RCO) method to extract relevant genes in gene expression data, which were afterward categorized with an RNN. On the Leukemia database, Small Blue Round Cell Tumor (SBRCT) dataset, and Lung Cancer Dataset, RCO-RNN achieved a 95% accuracy rate. Majji et al. [105] presented a novel technique for automatic cancer prediction, referred to as JayaALO-based DeepRNN, which employed Jaya ant lion optimization (ALO) in an RNN model. The approach was validated using four datasets, namely AP Colon Kidney, AP Breast Ovary, AP Breast Colon, and AP Breast Kidney dataset, and achieved the maximum accuracy of 95.97%. Suresh et al. [106] designed an approach for interpreting genome sequencing with the bat sonar algorithm and LSTM model for disease detection. LSTM recurrent networks were frequently used in other related works to find associated genes for tumor diagnosis, breast cancer detection, identify cancerous cells from normal cells, and biological entity recognition [107,108,109,110]. Zhao et al. [111] developed an RNN model to identify the transcriptional target factor. The memetic technique was used in [112] to learn RNN parameters, while LASSO-RNN was used to rebuild gene regulatory networks (GRNs). A summary of recent related works based on RNN is provided in Table 8. 

RNN models have multiple advantages in gene expression analysis, as they improve the efficiency by enabling the model to identify and retain sequential feature information [113]. Additionally, these networks can adapt to the dynamics of uncertain systems, for instance, because the significance of genetic data may alter over time. RNNs also have some disadvantages for gene expression analysis, such as increased processing time compared to CNNs and other related methods, resulting in slower and more complex training procedures and a reduced ability to capture dependencies in longer genomic sequences compared to GNN and TNN [114,115].

#### 4.2.3. Convolutional Neural Networks (CNN)

Convolution Neural Networks (CNN) are deep learning architectures initially designed primarily for image analysis and processing. CNN employ convolutional filters to automatically learn spatial feature hierarchies in input data. The network architectures use a combination of stacked convolutional and pooling layers (additional regularization layers are frequently used, such as Batch Normalization or Dropout layers) [116]. In gene expression analysis, Xiao et al. [117] presented a CNN-based ensemble method, which was applied to three public RNA-Seq datasets of three kinds of cancers, including Lung Adenocarcinoma, Stomach Adenocarcinoma, and Breast Invasive Carcinoma, and attained a precision of 98%. In several related research works [18,118], the authors employed CNN models to classify tumor types by embedding the high-dimensional RNA-Seq data into 2D images. Accordingly, the lightweight CNN architecture for breast cancer classification using gene expression data transformed into 2D images proposed by Elbashir et al. [18] achieved a precision of 98.76%. Three CNN models (1D-CNN, 2D-Vanilla-CNN, and 2D-Hybrid-CNN) were trained and tested using gene expression profiles from 10,340 samples of 33 different cancer types. The authors fed 713 normal samples that matched 23 TCGA cancer types into a 1D-CNN model to examine the effects of tissues of origin. This model achieved the best performance for predicting breast cancer subtypes compared to other models, with a precision of 88.42% [119].

CNN using multi-dimensional 1D, 2D, and 3D convolutional models has been designed for analysis of gene expression data. One-dimensional convolutions were applied to sequences of genomic data and were proven suitable for learning sequential patterns. Two-dimensional convolutions have been designed for processing gene expression data that are first transformed into an image format. Gene expression images are typically created by directly mapping gene expression values to a predetermined palette of colors and utilizing domain-specific data to identify the location of each gene in the images. The main disadvantage of this method is the loss of information that results from utilizing discrete sets of colors instead of the original continuous expression values to calculate the values of the image pixels. Another approach divides the process of creating a gene expression image into two parts that follow one another. First, a biological functional hierarchy represented by a tree-shaped structure is transformed into a functional hierarchy image template. The gene locations are defined by following a certain biological standard. Afterward, gene expression images are created by mapping the expression values to the positions of the genes in the image template. This results in a final set of images where each pixel represents the continuous gene expression value of the corresponding gene, preventing the information loss that results from converting continuous gene expression values into a set of discrete colors [120]. CNN models have generally achieved better performance for gene expression analysis in comparison to RNN (see Table 9) because of the ability to evaluate large amounts of genetic data more quickly, effectively, and accurately, by extracting relevant information from both local and global level features in gene expression data [36,121].

#### 4.2.4. Graph Neural Networks (GNN)

Graph neural networks (GNN) [124] are a deep learning architecture developed for performing inference on data represented by graphs with vertices (nodes) and edges. A graph network propagates data features through the graph nodes to learn contextualized features via a statistical model for analyzing pairwise connections between objects and entities. It aims to create precise state embedding vectors, where the state of the nodes is continuously updated with the information dissemination mechanism on the graph. GNN models follow a neighborhood aggregation scheme, where the representation vector of a node is computed by recursively aggregating and transforming the representation vectors of its neighboring nodes. For biological networks, graph nodes are frequently genes, transcripts, or proteins, whereas graph edges tend to represent experimentally determined similarities or functional linkages between them. The generation of network graphs [125] from gene expression data frequently uses correlation coefficients to measure the similarity between gene expression profiles derived from the range of analyzed samples. For instance, pairwise Pearson correlation coefficients calculated for every set on the array and above a predefined threshold have been used to define edges between genes (nodes) network graphs.

GNN models have been designed for the analysis of multi-omics pan-cancer data such as gene expression profile, DNA methylation, gene mutation rates, copy number variation, exon expression, and clinical data, with an emphasis on predicting various types of cancers [25]. Pfeifer et al. [126] introduced a unique explainable GNN-based framework for cancer subnetwork discovery. The protein–protein interaction (PPI) network topology of each patient is employed, where the nodes are enriched with multi-omics data from DNA methylation and gene expression. The proposed GNN explainer offers model-wide explanations for enhanced illness subnetwork detection. Similarly, other studies have employed GNN to forecast cancer types and find cancer-specific indicators using multi-omics data with PPI networks [127]. Zhou et al. [128] employed gene–gene interaction networks for cancer prediction in multi-dimensional omics datasets using a graph convolutional network (GCN). In order to improve the diagnostic performance of graph-based techniques for cancer grading, the authors used a contour-aware information aggregation network (CIA-Net) with nuclear masks to extract nuclear shape and appearance features. A gated graph attention network (GGAT) [26] was designed to extract the underlying semantic information in graph-structured data where the graph describes the relationship between genes and their associated molecular functions. The authors used a gating mechanism (GM) that interacts with the attention mechanism (AM) to overcome the limitation of one-hop neighborhood reasoning (i.e., every node embedding contains information about the features of its immediate graph neighbors, which can be reached by a path of length one in the graph). In another related study, the authors achieved an accuracy of 95.66% by gaining knowledge for distinguishing between the relative importance of the nodes in each surrounding node [129].

GNN models have multiple advantages for processing gene expression data due to the intrinsic learnable properties of propagating and aggregating attributes that capture relationships across the entire cell-cell graph. Hence, the learned graph embeddings can be treated as high-order representations of cell–cell relationships in RNA-Seq data in the context of graph topology. GNNs can also effectively aggregate detailed relationships between similar cells using a bottom-up approach, and they can use prior domain knowledge in gene regulation to direct the imputation of missing data [130]. Furthermore, GNN models have the ability to combine the strengths of heuristic skeletons. By incorporating topological neighbor propagation throughout the entire gene network, GNNs offer the means to build gene regulatory networks (GRN) and improve the generalization capability [131]. One of the shortcomings of GNN is the sensitivity to noisy data when building graph structures [132].

#### 4.2.5. Transformer Neural Networks (TNN)

TNN models employ network architectures that are based on the multi-head self-attention mechanism, which allows capturing long-range dependencies between items in a sequence [133]. It is a state-of-the-art approach for processing sequential data, such as time series data, genomic sequences, acoustic signals, or natural language data. TNN is an appealing network choice for gene expression analysis due to the ability to jointly attend to information from different representation subspaces at different positions in genomic data. Gene transformer [27] employs multi-head self-attention modules to address the complexity of high-dimensional gene expression by recognizing relevant biomarkers across multiple cancer subtypes. The multi-omic transformer adopted the transformer architecture proposed by Osseni et al. [28] to discriminate complex phenotypes (cancer types) based on four omics data types: transcriptomics (mRNA and miRNA), epigenomics (DNA methylation), copy number variations (CNVs), and proteomics. Lv et al. [134] introduced a transformer-based fusion network integrating pathological images and genomic data (PG-TFNet) for cancer survival analysis. The transformer-based feature fusion module allowed researchers to leverage the intra-modality relationships between patches in multiple fields of view in multi-scale pathological slides.

TNN models have demonstrated increased robustness in comparison to CNN and RNN and exhibited competitive performance on benchmarks with various data formats. The self-attention mechanism allows one to utilize contextual information for any location in the input sequence and capture long-range dependencies in comparison to CNN, and permits higher parallelization compared to RNN [135]. Among the drawbacks of TNNs include the requirement for large amounts of data, and hence their performance can be inferior to other NN models for genetic data with a lower number of input samples [136].

### 4.3. Transfer Learning

Transfer learning [137] aims to enhance the performance of downstream models by transferring information from different (but related) source domains. In order to use knowledge representation of feature maps to untrained cancer datasets, Kakati et al. [138] employed transfer learning to a CNN model called DEGnext, to predict the significant up-regulated (UR) and down-regulated (DR) genes from gene expression data received from The Cancer Genome Atlas database. Das et al. [24] utilized spectrogram images of digital DNA sequences to perform transfer learning for automated liver cancer gene recognition using 2D CNN models. In their suggested method, DNA sequences are digitally represented using entropy-based, EIIP, and integer-mapping approaches. Zhang et al. [139] fine-tuned a convolutional LSTM network (CLSTM) by transfer learning to model the temporal genetic information of cancer genes in dynamic contrast-enhanced magnetic resonance imaging (DCE-MRI). Kandaswamy et al. [140] introduced a deep transfer learning (DTL) framework built upon individual cell information without employing any type of profiling or reduction methods with extracted cell features, which sped up the process by 30% and improved the performance. The characteristics of related transfer learning approaches are shown in Table 10.

### 4.4. Pathway Analysis

Pathway analysis is a widely used technique for extracting biological meaning from high-throughput gene expression data. Existing methods primarily focus on determining which pathway or pathways may have been disrupted because of differential gene expression patterns [142]. For instance, the adipocytokine signaling pathway was used to effectively distinguish breast cancer from colon and stomach tumors [143]. These methods are critical for developing effective bioinformatics techniques that enable researchers to understand the genes and pathway routes altered in various cancer types and find potential treatments. Over the last two decades, many pathway analysis methods have been proposed, all of which can be divided into three categories (i.e., generations) based on their timeline and employed strategy. The first two generations are referred to as over-representation analysis (ORA) and functional class scoring (FCS) [144], and they use pathways as gene sets. Topology-based (TB) pathway analysis [145] is the third generation, which incorporates the pathway topology into the model for improved performance [146]. In TB pathway analysis, the signaling pathways method uses two types of information to calculate a pathway’s impact, which includes differentially expressed (DE) genes in a particular pathway and additional biological information related to the position and degree of change in all DE genes expression, the relationships between genes as specified by the pathway, and the nature of interactions. Biological pathway databases, such as KEGG [147] and Reactome [148], utilize years of curated knowledge to annotate the positions and interactions of genes in a pathway.

## 5. Future Directions

This section discusses future directions that may potentially advance the research on ML-based gene expression analysis.

One avenue of future research is to consider additional types of input features with existing learning algorithms because the full impact of gene expression cannot be represented by the genetic sequence alone. Specifically, DNA methylations and mutations are possible types of features that can be utilized in cancer classification. DNA methylations can occur at CpG dinucleotides as well as in non-CpG sites. The CpG is used to differentiate between the CG base-pairing of cytosine and guanine from the single-stranded linear sequence. DNA methylation is linked to the normal developmental process and the observable change during the pathological processes. The pathological processes include DNA repair genes and the gene-silencing of tumor suppressors. Therefore, integrating methylations and mutations with RNA-Seq data can produce features that positively impact tumor classification.Along with selecting the feature type that can contribute significantly to enhancing the performance of ML methods, the design of the computational algorithm is also essential. In this regard, researchers can focus on innovative techniques that can perform efficiently on gold-standard datasets, such as the unique molecular identifier (UMI), which has experimentally proven reference genes. Such studies can allow researchers to conduct an experimental comparison of single-cell methods. Also, research studies can validate the performance of algorithms on single-cell sequencing protocols such as SMART-Seq, Cel-Seqs, and droplets.Identifying cancer-related biomarkers can be an important future direction where researchers can investigate methodologies for identifying biomarkers related to each cancer type. For instance, the methods listed for IntPath [149] and others [150] can help in conducting functional pathway analysis of related genes for the cancer types. Given a 2D image, DL methods can be used to extract promising features from images, which can assist in identifying cancer-specific biomarkers.GNN can also be designed to support the integration of single-cell multi-omics data by implementing heterogeneous graphs. Such data can include Droplet scRNA-Seq [151] and the intra-modality of Smart-Seq2. Cell-type-specific gene regulatory mechanisms can be elucidated using scGNN, especially when integrating scATAC-Seq and scRNA-Seq data. Additionally, T cell ancestries can be identified uniquely by the T cell receptor repertoires. The unique identification of T cells is important because it can improve the performance of prediction methods regarding cell–cell interactions. scGNN can facilitate building connections between diverse experiments, sequencing technologies, and data modalities.It is also essential to place emphasis on the design of interpretable ML models that help to understand the decision-making process by the employed computational methods, and provide explanations about the cases where the models might fail. Interpretable and explainable models that highlight the local and global properties of ML models based on counterfactuals or feature attribution should receive increased attention in this area.Recently, more studies have considered the analogous genomic probing of pre-malignant lesions in the genomic analysis of various cancers, undertaken as part of The Cancer Genome Atlas (TCGA) project. In addition to genomics, cancer prevention strategies in the future will incorporate a variety of new modalities, including imaging, proteomic, metabolomic, glycemic, and epigenetic, to identify and validate surrogate biomarkers for cancer gene prevention trials. In conjunction with preclinical and clinical studies, these modalities can help establish new biomarkers to improve cancer patients’ treatment. Toxicologists, pathologists, and clinicians involved in early-phase clinical studies may be able to use these novel validated biomarkers for diagnosis, treatment, and cancer patient monitoring.Multidomain genomic data analysis is another important avenue to study feature selection and extraction, and for downstream analysis. Multimodal and multitask ML methods based on early and late fusion may provide improved performance compared to existing methods.Differentiating clinically similar cancers can be challenging, and focusing on genomic and transcriptomic variations may prove beneficial. The omics data describe details on various methods available for ovarian and different types of cancers and renal cell carcinoma for identifying key genes and pathways that might assist in proposing diagnostic and prognostic predictions. Optical genome mapping and structural variant analysis (at a region of DNA, also known as copy number variants, which can contain inversions, balanced translocations, or genomic imbalances) may be applied to various cancer datasets for improved prognosis and treatment [152].Further understanding of circRNA localization, transportation, and degradation in live cells, a completed circRNA interactome, and single-cell profiling are important topics in this field that may prove helpful for cancer gene prediction [153].

A summary of future perspectives is provided in Table 11.

## 6. Conclusions

Recent advances in deep learning-based approaches for processing complex high-dimensional data offer tremendous potential for pattern recognition and predictive analytics of multi-omics data. This study overviews the progress in the application of both traditional machine learning methods and deep learning methods for gene expression analysis using RNA-sequencing and DNA microarray data for cancer detection. The manuscript presents a brief overview of the data collection methods for gene expression analysis and lists the pertinent commonly used datasets for supervised machine learning. A taxonomy of the techniques for feature engineering and data preprocessing is also provided, as an important component of gene expression analysis. ML-based methods for gene expression analysis are presented next, with a focus on deep learning-based approaches, due to their comparative advantages for gene expression analysis. Prior works that employ neural networks with popular architectures are covered, including multi-layer perceptrons, as well as convolutional, recurrent, graph, and transformer networks. The use of deep learning methods in cancer classification using RNA-Seq data has shown promising results, with several studies reporting high accuracy in classifying different types of cancer. We expect that future research in this area will address the current challenges of generalizability, robustness, and explainability of the results, and will lead to enhanced cancer diagnosis and improved healthcare outcomes. The paper concludes with an outline of promising future directions for cancer classification using gene expression analysis.

The main contributions of this study are in the provision of a comprehensive review of recent research works for cancer classification using gene expression analysis, covering feature engineering techniques, datasets for gene expression analysis, and applications of traditional and deep learning ML methods. This study overviews methods based on recent neural network architectures—such as graph and transformer networks—that have not been covered in published reviews, as well as having a focus on RNA-Seq methods as the dominant data format in recent works.

## Figures and Tables

**Figure 1 bioengineering-10-00173-f001:**
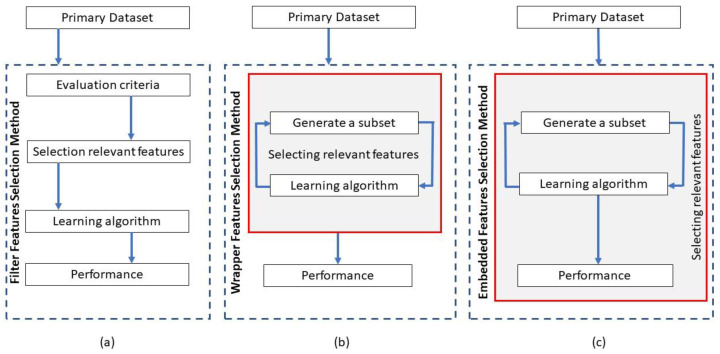
Information flow in the three primary categories of feature engineering methods: (**a**) filter, (**b**) wrapper, and (**c**) embedded methods [62].

**Table 1 bioengineering-10-00173-t001:** List of previous review papers for gene expression analysis.

Reference	Conventional ML Approaches	Feature Engineering	DL Approaches	Microarray Data	RNA-Seq Data
Sathe et al., 2019 [29]	No	No	RNN and CNN	Yes	Yes
Koumakis et al., 2020 [30]	No	No	RNN and CNN	Yes	Yes
Zhu et al., 2020 [31]	No	No	MLPNN, RNN, and CNN	Yes	Yes
Gunavathi et al., 2020 [32]	No	No	CNN	Yes	Yes
Tabares et al., 2020 [33]	Yes	No	MLP and CNN	Yes	No
Bhonde et al., 2021 [12]	No	No	MLPNN, RNN, and CNN	Yes	Yes
Mazlan et al., 2021 [34]	Yes	Yes	CNN	Yes	Yes
Karim et al., 2021 [35]	Yes	Yes	MLPNN, RNN, and CNN	Yes	Yes
Thakur et al., 2021 [36]	Yes	No	CNN	Yes	Yes
Montesinos-López et al., 2021 [37]	No	No	MLPNN, RNN, and CNN	Yes	Yes
Bhandari et al., 2022 [38]	Yes	Yes	MLPNN, RNN, and CNN	Yes	No
Khalsan et al., 2022 [39]	Yes	No	MLPNN, RNN, and CNN	Yes	Yes
Alhenawi et al., 2022 [40]	No	Yes	No	Yes	No

**Table 2 bioengineering-10-00173-t002:** Comparison between microarray and RNA-Seq data.

Characteristics	Microarray Data	RNA-Seq Data
Gene Discovery	No	Yes
Different Isoform	No	Yes
High Resolution	No	Yes
Background Noise	Yes	No
High Cost	Yes	No
Rare/New Transcript	No	Yes
Noncoding RNA	No	Yes

**Table 3 bioengineering-10-00173-t003:** Datasets for gene expression analysis.

Reference	Classification Task	Type of Cancer and Data Source	Number of Samples	Type of Data
Mohammed et al., 2021 [45]	Multiclass Classification (5 types of cancers)	Breast Cancer (BRCA), Colon adenocarcinoma (COAD), Lung adenocarcinoma (LUAD), Ovarian (OV), and Thyroid Cancer (THCA) from Pan-Cancer Atlas	2166	RNA-Seq
Li et al., 2022 [46]	Binary Classification	Kidney Renal clear cell carcinoma (KIRC) from The Cancer Genome Atlas (TCGA)	945	RNA-Seq
Zhang et al., 2022 [47]	Binary Classification	Liver Hepatocellular Carcinoma (LIHC) from The Cancer Genome Atlas (TCGA)	424	RNA-Seq
Coleto-Alcudia et al., 2022 [48]	Binary Classification	Breast Cancer (BC) from The Cancer Genome Atlas (TCGA)	1178	RNA-Seq
Abdelwahab et al., 2022 [49]	Binary Classification	Lung Adenocarcinoma (LUAD) from The Cancer Genome Atlas (TCGA)	549	RNA-Seq
Ke et al., 2022 [50]	Multiclass Classification (33 types of cancers)	33 Types of Cancer from The Cancer Genome Atlas (TCGA)	10,528	RNA-Seq
Divate et al., 2022 [51]	Multiclass Classification (39 types of cancers)	39 Types of Cancer from The Cancer Genome Atlas (TCGA)	14,237	RNA-Seq
Houssein et al., 2021 [52]	Binary Classification	Leukemia from the GEO (Gene Expression Omnibus)	72	Microarray
Hira et al., 2021 [53]	Multiclass Classification (18 types of cancers)	18 Types of Cancer from GEO (Gene Expression Omnibus)	2096	Microarray
Vaiyapuri et al., 2022 [54]	Binary Classification	Ovarian Cancer from the GEO (Gene Expression Omnibus)	253	Microarray
Lin Ke et al., 2022 [55]	Binary Classification	Lung Cancer from the GEO (Gene Expression Omnibus)	181	Microarray
Deng et al., 2022 [56]	Binary Classification	Myeloma from the GEO (Gene Expression Omnibus)	173	Microarray
Rostami et al., 2022 [57]	Binary Classification	Prostate Cancer from the GEO (Gene Expression Omnibus)	102	Microarray
Xie et al., 2022 [58]	Binary Classification	Colon Cancer from the GEO (Gene Expression Omnibus)	62	Microarray

**Table 4 bioengineering-10-00173-t004:** Feature selection methods in gene expression analysis.

Reference	Feature Selection Method	Feature Selection Algorithm	Dataset Type	Accuracy (%)
Park et al., 2019 [65]	Filter Methods	Artificial Neural Network (ANN)	RNA-Seq	90.71%
García-Díaz et al., 2019 [66]	Filter Methods	Grouping Genetic Algorithm	RNA-Seq	98.81%
Wu and Hicks, 2021 [67]	Filter Methods	K-nearest neighbor (kNN) Naïve Bayes (NGB) Decision trees (DT) Support Vector Machines (SVM)	RNA-Seq	87% 85% 87% 90%
Chen and Dhahbi, 2021 [68]	Filter Methods	RF	RNA-Seq	90%
Liu and Yao, 2022 [69]	Filter Methods	Deep Neural Network (DNN)	RNA-Seq	99%
Gakii et al., 2022 [70]	Filter Methods	Multilayer Perceptron Sequential Minimal Optimization Naive Bayes Classifier	NSCLC RNA-Seq	100% 96.42% 98.59%
Mahin et al., 2022 [71]	Filter Methods	k-Nearest Neighbor	RNA-Seq	100%
Li et al., 2017 [72]	Wrapper Methods	Genetic Algorithm /k-Nearest Neighbor	RNA-Seq	90%
Zhang et al., 2018 [73]	Wrapper Methods	SVM-RFE-GS SVM-RFE-PSO SVM-RFE-GA RFFS-GS	RNA-Seq	91% 91.68% 91.34% 92.19%
Simsek et al., 2020 [74]	Wrapper Methods	RF Artificial Neural Networks DL Model (RMSProp)	RNA-Seq	91.83% 89.22% 95.15%
Al-Obeidat et al., 2021 [75]	Wrapper Methods	BABC-SVM	RNA-Seq	97.41% 97.35% 98.50% 95.86%
Liu et al., 2022 [76]	Wrapper Methods	Random Forest	RNA-Seq	99.68%
Al Abir et al., 2022 [77]	Wrapper Methods	Support Vector Machines (SVM) SVM-RFE	RNA-Seq	99.93%
Kong and Yu, 2018 [78]	Embedded Methods	Graph-Embedded Deep Feedforward Networks (GEDFN)	BRCA RNA-Seq	94.50%
Jiang et al., 2020 [79]	Embedded Methods	Bayesian Robit regression with Hyper-LASSO (BayesHL)	RNA-Seq	N/A
Zhang and Liu, 2021 [80]	Embedded Methods	Robust biomarker discovery framework	RNA-Seq	97% 98% 99% 98% 99% 98%
Abdelwahab et al., 2022 [49]	Embedded Methods	Recursive Feature Elimination (RFE) + SVM	RNA-Seq	94%
Coleto-Alcudia et al., 2022 [48]	Embedded Methods	Filtering + SVM	RNA-Seq	93%

**Table 5 bioengineering-10-00173-t005:** Characteristics of the categories of feature engineering methods for gene expression analysis.

Feature Selection	Filter Methods	Wrapper Methods	Embedded Methods
Pros	Univariate	Deterministic	Interacts with the classifier in a complex way. Models feature dependencies. Reduced computational complexity than wrapper methods.
	Fast and scalable to large datasets. Independent of the classifier.	Interacts with the classifier in a simple way. Models feature dependencies. Takes less time to compute than randomized methods.	
	Multivariate	Randomized	
	Models feature dependencies. Independent of the classifier. Reduced computational complexity than wrapper methods.	Interacts with the classifier. Models feature dependencies. Less prone to the local feature interaction problem.	
Cons	Univariate	Deterministic	Classifier-dependent selection.
	Ignores feature dependencies. Ignores interaction with the classifier.	Risk of overfitting. More prone than randomized algorithms to the local feature interaction problem. Classifier-dependent selection.	
	Multivariate	Randomized	
	Slower and less scalable than univariate techniques. Ignores interaction with the classifier.	Computationally intensive. Models feature dependencies. Classifier-dependent selection. Higher risk of overfitting than deterministic algorithms.	

**Table 6 bioengineering-10-00173-t006:** Traditional ML-based methods for gene expression analysis.

Reference	Dataset	Algorithm	Dataset Type	Performance
Segal et al., 2003 [86]	Cancer	SVM	Gene Expression Data	Accuracy: 98.5%
Hijazi et al., 2013 [88]	Mixed-Lineage Leukemia (MLL)	SVM Linear	Gene Expression Data	Accuracy: 99.89%
Ram et al., 2017 [87]	Colon Cancer	RF	Microarray Data	Accuracy: 87.39%
Zhang et al., 2018 [73]	Breast Cancer	SVM-RFE-PSO	Gene Expression Data	Accuracy: 81.54%
Yuan et al., 2020 [89]	Tumor-educated platelets	Evolutionary Programming-trained SVM	Gene Expression Data	Accuracy: 95.93%
Yuan et al., 2020 [90]	Lung adenocarcinoma (AC) and lung squamous cell cancer (SCC)	RF	Gene Expression Data	Accuracy: 94.9%
		RF	Gene Expression Data	Accuracy: 93.3%
		SVM	Gene Expression Data	Accuracy: 94.7%
Abdulqader et al., 2020 [91]	Lymphoma	kNN	Microarray Data	Accuracy: 94.7%
	Lymphoma	NB	Microarray Data	Accuracy: 74.83%

**Table 7 bioengineering-10-00173-t007:** Deep learning-based MLP methods for gene expression analysis.

Reference	Dataset	Type of Cancer	Algorithm	Dataset Type	Performance
Zhang et al., 2018 [95]	NCBI GEO database	Breast	DNN [AdaBoost algorithm (PCA-AE-Ada)]	Gene Expression Data	ROC-AUC: 0.714%
Gao et al., 2019 [96]	Breast Cancer	Breast	DeepCC	Gene Expression Data	Accuracy: 89%
Lai et al., 2020 [94]	Lung Adenocarcinoma	Lung	DNN [four hidden layers, with the rectified linear unit (ReLU)]	Gene Expression Data	ROC-AUC: 0.8163, Accuracy: 75.44%
Chandrasekar et al., 2020 [97]	Microarray Dataset	Cancer (heterogeneous disease)	DNN	Gene Expression Data	Accuracy: 72.5%
Laplante et al., 2020 [98]	TCGA	Tumor	DNN	Gene Expression Data	Accuracy: 96.9%
Azad et al., 2021 [99]	Breast Cancer Wisconsin (Original) Dataset (WBCD)	Breast Cancer	Intelligent Ensemble Classification method based on Multi-Layer Perceptron neural network (IEC-MLP)	Gene Expression Data	Accuracy: 98.74%
Alshareef et al., 2022 [100]	Patient Databases (PubMed, CENTRAL, EMBASE, OASIS, and CNKI)	Prostate Cancer Detection	DNN [CIWO-based F.S.]	Gene Expression Data	Accuracy: 96.21%

**Table 8 bioengineering-10-00173-t008:** Deep learning-based RNN methods for gene expression analysis.

Reference	Dataset	Type of Cancer	Algorithm	Dataset Type	Performance
Sahin et al., 2019 [103]	Colon, lung, and prostate datasets from gene expression profile (GEP) datasets	Lung Lymphoma Leukaemia Colon SRBCT Prostate	RNN [LSTM-AIRS]	Microarray data	Accuracy: 89.6% 88.3% 85.3% 84.7% 77.6% 75.7%
Zhao et al., 2019 [111]	Data1endoderm (PrE) Cells. Data2: Mouse embryonic fibroblast (MEF) cells. Data3: Definitive endoderm (DE) cells.	Classification	RNN	Gene Expression Data	ROC-AUC Data1: 0.620 Data2: 0.587 Data3: 0.578
Liu et al., 2020 [112]	Benchmark Dataset: DREAM3 and DREAM4	Classification	MALASSRNN-GRN	Microarray data	ROC-AUC (node = 10, Density = 20%, 40%) 0.6351 0.7188
Aher et al., 2021 [104]	Leukaemia datasets (Leukemia data 2017), SRBCT dataset (SBRCT dataset 2020). SRBCT dataset (SBRCT dataset 2020).	Leukaemia SRBCT Lung	RNN [Rider Chicken Optimization algorithm] (RCO-RNN)	Gene Expression Data	Accuracy: 94.5% 94.0% 95.0%
Majji et al., 2021 [105]	AP_Colon_Kidney dataset 2020 AP_Breast_Ovarydataset 2020 AP_Breast_Colon dataset 2020 AP_Breast_Kidney dataset 2020	Colon Cancer Kidney Cancer Breast Cancer Ovary Cancer	JayaALO-based DeepRNN	Gene Expression Data	Accuracy: 95.27% 95.97% 95.97% 95.27%
Suresh et al., 2021 [106]	Acute myeloid (AML) and acute lymphoblastic leukemia (ALL)	Acute myeloid (AML) and acute lymphoblastic leukemia (ALL)	LSTM and Bat sonar Algorithm	Gene Expression Data (via DNA microarray)	Accuracy: 86.35%

**Table 9 bioengineering-10-00173-t009:** Deep learning-based CNN methods for gene expression analysis.

Reference	Dataset	Type of Cancer	Algorithm	Dataset Type	Performance
Xiao et al., 2018 [117]	From TCGA LUAD dataset STAD dataset BRCA dataset	LUAD STAD BRCA	Deep learning-based multi-model ensemble method	RNA-seq data	Accuracy: 95.60% 94.63% 94.62%
Lyu et al., 2018 [118]	33 tumor types in Pan-Cancer Atlas	33 tumor types.	CNN	RNA-Seq data	Accuracy: 95.59%
de Guia, et al., 2019 [122]	TCGA	33 cohorts of cancer types	CNN	RNA-Seq data	Accuracy: 95.65%
Elbashir et al., 2019 [18]	TCGA	Breast Cancer	lightweight CNN	RNA-Seq data	Accuracy: 98.76%
Mostavi et al., 2020 [119]	From TCGA: 33 cancer types 23 normal tissues.	33 cancer types and 23 normal tissues.	1D-CNN 2D-Vanilla-CNN 2D-Hybrid-CNN	RNA-Seq data	Accuracy: 95.7% 92.5% 95.7%
Khalifa et al., 2020 [123]	Tumor Gene Expression for five separate cancer types	KIRC BRCA LUSC LUAD UCEC	Binary Particle Swarm Optimization–Decision Tree (BPSO—DT) and CNN.	RNA-Seq data	Accuracy: 98.20% 98.30% 97.7% 84.8% 96.40%

**Table 10 bioengineering-10-00173-t010:** Deep-learning-based Transfer Learning methods for gene expression analysis.

Reference	Type of Cancer	Algorithm	Dataset Type	Performance
Sevakula et al., 2019 [141]	AP_Omentum_Lung (OL) AP_Omentum_Uterus (OU) AP_Colon_Omentum(CO) AP_Ovary_Uterus (OvU) AP_Endometrium_Uterus (EU) AP_Endometrium_Ovary (EOv) AP_Omentum_Ovary(OOv)	Transfer Learning	Gene Expression Data	Accuracy: 98.30% 97.89% 97.08% 95.60% 94.85% 93.82% 84.80%
Lopez-Garcia et al., 2020 [120]	Lung Cancer	Transfer Learning + CNN	Gene Expression Data	Accuracy: 73.26%
Zhang et al., 2021 [139]	Breast cancer	Transfer Learning + CNN + CLSTM	Breast cancer molecular subtypes on MRI	Accuracy: CNN with transfer learning = 90% CLTSM with transfer learning = 93%
Kakati et al., 2022 [138]	Breast cancer and uterine cancer	Transfer Learning + CNN	Gene Expression Data	ROC-AUC: 88–99%
Das et al., 2022 [24]	Liver cancer	Deep Transfer Learning	Gene Sequences	Accuracy: 1D CNN model = 80.36% 2D CNN model = 98.86%

**Table 11 bioengineering-10-00173-t011:** Future directions in ML-based methods using gene expression data.

Future Perspectives	
New types of data features	Introducing additional input features, such as DNA methylations and mutations, can improve the discriminative performance of existing learning algorithms.
2. Innovation in computational algorithms	The design of novel computational algorithms and novel benchmarking approaches is important for advancing gene expression analysis.
3. Improved cancer-related biomarkers	Investigate methods for identifying biomarkers specific to each form of cancer.
4. Integration of single-cell multi-omics data with graph networks	GNN architectures can support the integration of single-cell multi-omics data by employing heterogeneous graphs.
5. Design interpretable and explainable approaches	Emphasize the adoption of interpretable ML models that help to understand the decision-making process and explain the reasons when ML models fail.
6. Cancer prevention strategies based on multiple data modalities	Combine a variety of new modalities, including imaging, proteomic, metabolomic, glycemic, and epigenetic data, to find and evaluate surrogate biomarkers for cancer gene prevention studies.
7. Design multi-modal and multi-task learning approaches	Multimodal and multitask ML methods based on early and late fusion have the potential to improve classification performance.

## Data Availability

Not applicable.

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
