# Peer review of "Machine Learning Methods for Cancer Classification Using Gene Expression Data: A Review"

_bioengineering, 2023, doi:10.3390/bioengineering10020173_

Round 1

Reviewer 1 Report

This study reviewed and explored the implementation of machine learning techniques in classifying the Caner Using Gene Expression. The current work served the aim of the bioengineering Journal. However, an in-depth analysis of the studies and how each method is constructed and evaluated must be conducted. The survey should show the studies with the added value base on meta-analysis and synthesis (no narrative review). The survey has to be theoretically and critically analyzed with sufficient background. Also, it must justify the paper selection and the aspects to rationalize your viewpoint. Besides, the technical writing has many grammatical errors, making it hard to read. The authors should justify the novelty of the proposed work as similar studies have been carried out in the existing literature.  

In addition to the following:

-Enhance the conclusion to present the findings and contributions of the work.

- I suggest changing the Machine learning methods instead of  Deep Learning-based Methods in the paper's title, as you mentioned in L14-15. 

-Enhance the presentation of tables. 

- Check and unify the reference in one style, such as [20 - 24], [30 – 39], [45 – 57], [ 61], [63 – 64], etc.

- Add a reference to support the text in L8-9.

- Describe and conclude the advantages and disadvantages of each mentioned method in the literature studies.

-What are the current research gaps in the studies mentioned in the literature survey, and how will this work fill them?

The research paper should be written in the third person's perspective; words such as "we", "our," etc., must be avoided.

- Avoid using many references together, such as L34: [2-4], L81: [16-18], L127: [18, 31, 32, 33, 34, 35], L234 [58-61], etc. You should classify the studies and write a proper paragraph bout each study or category.

- Abbreviations must be written in the complete form where they are first used, such as 

 DNA, RNA, cDNA, etc. Check the main text and edit for the same.

 - Enhance the Figure resolution, such as in Fig. 1, 

-Too-long sentences make the meaning unclear. Consider breaking it into multiple sentences—for example, L32-L35;  L35-L38;  L40-L43; L59-L62; etc.

-Many grammatical or spelling errors make the meaning unclear, and sentence construction errors need proofreading. Improve the English language, redaction, and punctuation in general. The manuscript should undergo editing before being submitted to the Journal again. 

The following are some examples:

L23:  gene expression data, caused by the large   …. Should be … gene expression data caused by a large

L31:  many cases they can result…. Should be … many cases, they can result

L36-37:  for early detection of cancer, as well as ….  Should be …   for the early detection of cancer, and … 

L40:  It involves transcription of DNA  …. Should be … It involves the transcription of DNA

L41:  followed by translation into  …. Should be … followed by a translation into

Author Response

We would like to thank the Editor and anonymous Reviewers for their valuable comments toward improving the manuscript. We have revised the manuscript thoroughly and taken all the suggestions of the Reviewers into careful consideration. 

Enclosed please find our responses to the Reviewers’ comments.

Response to Reviewer # 1

Comment 1: This study reviewed and explored the implementation of machine learning techniques in classifying Cancer Using Gene Expression. The current work served the aim of the bioengineering Journal. However, an in-depth analysis of the studies and how each method is constructed and evaluated must be conducted. The survey should show the studies with the added value base on meta-analysis and synthesis (no narrative review).

Response: Thank you for the many constructive comments. They were of great help for improving the manuscript. To address this comment we added additional details in the Introduction and Methods sections. We believe that the review paper is already quite long (about 30 pages with single spacing), and adding detailed information about the construction of each method will make it way too long. If it is needed, we would be glad to add the requested information in an Appendix section. Regarding the added value besides the narrative review, we believe that the tables in the manuscript serve this purpose. The tables list all recent papers, and provide information about the used data, methods, and performance of the reviewed papers. The tables can help readers in obtaining a picture of this field, without the need to read through the narrative text.

Comment 2: The survey has to be theoretically and critically analyzed with sufficient background. Also, it must justify the paper selection and the aspects to rationalize your viewpoint.
Response: We have updated the manuscript, and in particular, we expanded the Introduction section significantly to address this suggestion. We have justified the paper selection based on their importance and relevance to the reviewed topic. As we explained, our goal was to provide an overview of ML and DL papers for gene expression analysis, and to overview feature engineering techniques and datasets for this task.

Comment 3: Besides, the technical writing has many grammatical errors, making it hard to read.
Response: Thank you, we have carefully proofread the entire text to improve the English language. We have corrected all grammatical and spelling errors in the manuscript.

Comment 4: The authors should justify the novelty of the proposed work as similar studies have been carried out in the existing literature.

Response: Thank you for the great comment, we revised parts of the Introduction section to justify the novelty of the proposed work. The main contributions are listed in the last paragraph of the Introduction. The contributions involve: a comprehensive review of both conventional ML and recent DL methods for gene expression analysis (e.g., GNN and TNN are recent DL architectures, and recent works on gene expression analysis with GNN and TNN are not covered in other reviews), related RNA-Seq methods are reviewed as the dominant data format for this task in recent years, and overview of feature engineering techniques and dataset for ML-based gene expression analysis (which are not reviewed in many related surveys). Also, Table 1 provides a list of covered topics in similar review papers, and it helps to identify the differences with this review.  

Comment 5: Enhance the conclusion to present the findings and contributions of the work?
Response: We have revised the Conclusion section according to the comment. We provided a brief explanation about the potential and challenges of ML-based gene expression analysis, and we added one paragraph that summarizes the contributions of this work.

Comment 6: I suggest changing the Machine learning methods instead of Deep Learning-based Methods in the paper's title, as you mentioned in L14-15?

Response: Thank you for the suggestion, we replaced “Deep Learning-based Methods” with “Machine Learning Methods” in the paper's title.

Comment 7: Enhance the presentation of tables.

Response: The presentation of the tables has been enhanced. We have updated all tables and ensured that there is consistency in the font size, left-right spacing, border settings, and other table properties.

Comment 8: Check and unify the reference in one style, such as [20 - 24], [30 – 39], [45 – 57], [61], [63 – 64], etc.?

Response: The references have been unified according to the journal style.

Comment 9: Add a reference to support the text in L8-9.

Response: A reference has been added in the Introduction section L32 to support the statement that is mentioned in L8-9 in the Abstract.

Comment 10: Describe and conclude the advantages and disadvantages of each mentioned method in literature studies?

Response: Thank you, we have listed the main advantages and disadvantages of each method in the Introduction section. In addition, in the Methods section, we have provided more detailed explanations regarding the advantages and disadvantages of the reviewed methods.

Comment 11: What are the current research gaps in the studies mentioned in the literature survey, and how will this work fill them?

Response: We updated the Introduction section with explanations about the current research gaps in the literature. We addressed this comment partially in the answer to Comment 4 above. Existing reviews mostly covered conventional ML approaches, and even the reviews that covered DL approaches focused on MLP, CNN, and RNN. We are not aware of review papers that cover applications of recent DL architectures, such as GNN and TNN for gene expression analysis. Also, we believe that this review provides comprehensive information regarding the feature engineering techniques, datasets, and relevant methods.

Comment 12: The research paper should be written in the third person's perspective; must words such as "we", "our," etc., be avoided?

Response: Thank you, we have corrected all cases in the manuscript that didn’t use the third person’s perspective.

Comment 13: Avoid using many references together, such as L34: [2-4], L81: [16-18], L127: [18, 31, 32, 33, 34, 35], L234 [58-61], etc. You should classify the studies and write a proper paragraph bout each study or category.

Response: We revised the usage of the references, and in the revised text we avoided using many references together.

Comment 14: Abbreviations must be written in the complete form where they are first used, such as DNA, RNA, cDNA, etc. Check the main text and edit for the same.

Response: All abbreviations have been updated and written in the complete form where they were first used.

Comment 15: Enhance the Figure resolution, such as in Fig. 1.

Response: We have enhanced the resolution of Figure 1 in the Feature Engineering section.

Comment 16: Too-long sentences make the meaning unclear. Consider breaking it into multiple sentences—for example, L32-L35; L35-L38; L40-L43; L59-L62; etc?

Response: Thank you, we have applied the suggestion, and many long sentences in the text have been broken into multiple shorter sentences, including the sentences listed in the comment.

Comment 17: Many grammatical or spelling errors make the meaning unclear, and sentence construction errors need proofreading. Improve the English language, redaction, and punctuation in general. The manuscript should undergo editing before being submitted to the Journal again.

Response: Thank you for the suggestion. We have carefully proofread the entire text to improve the English language. We have corrected all grammatical and spelling errors in the manuscript.

Reviewer 2 Report

I believe this review paper is well documented and is appropriate to fit the journal (Bioengineering). My suggestions to improve this review article are as follows:

-  In Table 6, if there are information about ROC-AUCs, please show them in the table.

- Regarding the future perspective section, it would be better to add a summarized figure to show clearly.

Author Response

We would like to thank the Editor and anonymous Reviewers for their valuable comments toward improving the manuscript. We have revised the manuscript thoroughly and taken all the suggestions of the Reviewers into careful consideration. 

Enclosed please find our responses to the Reviewers’ comments.

Response to Reviewer # 2

 Comment 1: I believe this review paper is well documented and is appropriate to fit the journal (Bioengineering).

Response: Thank you for your encouraging comment.

Comment 2: In Table 6, if there are information about ROC-AUCs, please show them in the table?

Response: The corresponding papers did not report the ROC-AUCs values, and that is the reason why we didn’t include them in Table 6. For the papers that reported ROC-AUCs, we inserted the values in Tables 7-10.

Comment 3: Regarding the future perspective section, it would be better to add a summarized figure to show clearly?

Response: Thank you for your constructive suggestion, we inserted Table 11 which summarizes the items in the Future Perspectives section.

Table 11. Future directions in ML-based methods using gene expression data.

Future Perspectives

1.        New types of data features

Introducing additional input features, such as DNA methylations and mutations, can improve the discriminative performance of existing learning algorithms.

2.        Innovation in computational algorithms

The design of novel computational algorithms and novel benchmarking approaches is important for advancing gene expression analysis.

3.        Improved cancer-related biomarkers

Investigate methods for identifying biomarkers specific to each form of cancer.

4.        Integration of single-cell multi-omics data with graph networks

GNN architectures can support the integration of single-cell multi-omics data by employing heterogeneous graphs.

5.        Design interpretable and explainable approaches

Emphasize the adoption of interpretable ML models that help to understand the decision-making process and explain the reasons when ML models fail.

6.        Cancer prevention strategies based on multiple data modalities

Combine a variety of new modalities, including imaging, proteomic, metabolomic, glycemic, and epigenetic data, to find and evaluate surrogate biomarkers for cancer gene prevention studies.

7.        Design multi-modal and multi-task learning approaches

Multimodal and multitask ML methods based on early and late fusion have the potential to improve classification performance.

Round 2

Reviewer 1 Report

The current revised version of the manuscript is enhanced to the level that can publish in the current form.